# An anciently diverged family of RNA binding proteins maintain correct splicing of a class of ultra-long exons through cryptic splice site repression

Chileleko Siachisumo[1†], Sara Luzzi[1*†], Saad Aldalaqan[1†], Gerald Hysenaj[1], Caroline Dalgliesh[1], Kathleen Cheung[2], Matthew R Gazzara[3], Ivaylo D Yonchev[4], Katherine James[5], Mahsa Kheirollahi Chadegani[1], Ingrid E Ehrmann[1], Graham R Smith[2], Simon J Cockell[2], Jennifer Munkley[1], Stuart A Wilson[4], Yoseph Barash[3], David J Elliott[1*]

[1]Biosciences Institute, Faculty of Medical Sciences, Newcastle University, Newcastle, United Kingdom; [2]Bioinformatics Support Unit, Faculty of Medical Sciences, Newcastle University, Newcastle, United Kingdom; [3]Department of Genetics, Perelman School of Medicine, University of Pennsylvania, Phildelphia, United States; [4]School of Biosciences, University of Sheffield, Sheffield, United Kingdom; [5]School of Computing, Newcastle University, Newcastle, United Kingdom

*For correspondence:
Sara.Luzzi@newcastle.ac.uk (SL);
David.Elliott@ncl.ac.uk (DJE)

†These authors contributed equally to this work

**Abstract** Previously, we showed that the germ cell-specific nuclear protein RBMXL2 represses cryptic splicing patterns during meiosis and is required for male fertility (Ehrmann et al., 2019). Here, we show that in somatic cells the similar yet ubiquitously expressed RBMX protein has similar functions. RBMX regulates a distinct class of exons that exceed the median human exon size. RBMX protein-RNA interactions are enriched within ultra-long exons, particularly within genes involved in genome stability, and repress the selection of cryptic splice sites that would compromise gene function. The *RBMX* gene is silenced during male meiosis due to sex chromosome inactivation. To test whether RBMXL2 might replace the function of RBMX during meiosis we induced expression of RBMXL2 and the more distantly related RBMY protein in somatic cells, finding each could rescue aberrant patterns of RNA processing caused by RBMX depletion. The C-terminal disordered domain of RBMXL2 is sufficient to rescue proper splicing control after RBMX depletion. Our data indicate that RBMX and RBMXL2 have parallel roles in somatic tissues and the germline that must have been conserved for at least 200 million years of mammalian evolution. We propose RBMX family proteins are particularly important for the splicing inclusion of some ultra-long exons with increased intrinsic susceptibility to cryptic splice site selection.

## eLife assessment

This **important** paper addresses the process by which cryptic splice sites that occur randomly in exons are ignored by the splicing machinery. Integrating state-of- the-art genome-wide approaches such as CLIP-seq with the study of individual examples, this study **convincingly** implicates members of RBMX family of RNA binding proteins in such cryptic splice site suppression and showcases its importance for the fidelity of expression of genes with very large exons.

## Introduction

Efficient gene expression in eukaryotes requires introns and exons to be correctly recognised by the spliceosome, the macromolecular machine that joins exons together. The spliceosome recognises short sequences called splice sites that are present at exon-intron junctions within precursor mRNAs. In higher organisms, there is some flexibility in splice site recognition, as most genes produce multiple mRNAs by alternative splicing. However, aberrant 'cryptic' splice sites that are weakly selected or totally ignored by the spliceosome occur frequently in the human genome and can function as decoys to interfere with gene expression (*Aldalaqan et al., 2022*; *Sibley et al., 2016*). Many cryptic splice sites are located amongst repetitive sequences within introns, where they are repressed by RNA-binding proteins belonging to the hnRNP family (*Attig et al., 2018*). However, cryptic splice sites can also be present within exons, and particularly can shorten long exons (by providing competing alternative splice sites) or cause the formation of exitrons (internal exon sequences that are removed as if they were introns) (*Marquez et al., 2015*).

The testis-specific nuclear RNA binding protein RBMXL2 was recently shown to repress cryptic splice site selection during meiosis, including within some ultra-long exons of genes involved in genome stability (*Ehrmann et al., 2019*). RBMXL2 is only expressed within the testis (*Aldalaqan et al., 2022*; *Ehrmann et al., 2019*), raising the question of how these same cryptic splice sites controlled by RBMXL2 are repressed in other parts of the body. Suggesting a possible answer to this question, RBMXL2 is part of an anciently diverged family of RNA-binding proteins. The *RBMXL2* gene evolved 65 million years ago following retro-transposition of the *RBMX* gene from the X chromosome to an autosome (*Ehrmann et al., 2019*). RBMX and RBMXL2 proteins (also known as hnRNP-G and hnRNP-GT) share 73% identity at the protein level and have the same modular structure comprising an N-terminal RNA recognition motif (RRM) and a C-terminal disordered region containing RGG repeats (*Figure 1A*). RBMX and RBMXL2 are also more distantly related to a gene called *RBMY* on the long arm of the Y chromosome that is deleted in some infertile men (with only ~37% identity between human RBMXL2 and RBMY) (*Elliott et al., 1997*; *Ma et al., 1993*). The role of RBMY in the germline is almost totally unknown, but RBMY protein has been implicated in splicing regulation (*Elliott et al., 2000*; *Venables et al., 2000*).

The location of *RBMX* and *RBMY* on the X and Y chromosomes has important implications for their expression patterns during meiosis. The X and Y chromosomes are inactivated during meiosis within a heterochromatic structure called the XY body (*Turner, 2015*; *Wang, 2004*). Meiosis is quite a long process, and to maintain cell viability during this extended period a number of autosomal retrogenes have evolved from essential X chromosome genes. These autosomal retrogenes are actively expressed during meiosis when the X chromosome is inactive. However, it is unknown whether RBMXL2 is functionally similar enough to RBMX to provide a direct replacement during meiosis, or whether RBMXL2 has evolved differently to control meiosis-specific patterns of expression. Suggesting somewhat different activities, RBMX was recently shown to activate exon splicing inclusion, via a mechanism involving binding to RNA through its C-terminal disordered domain facilitated by recognition of m6A residues and RNA polymerase II pausing (*Liu et al., 2017*; *Zhou et al., 2019*).

Here, we have used iCLIP and RNA-seq to analyse the binding characteristics and RNA processing targets of human RBMX. We identify a novel class of RBMX-dependent ultra-long exons connected to genome stability and transcriptional control, and find that RBMX, RBMXL2, and RBMY paralogs have closely related functional activity in repressing cryptic splice site selection. Our data reveal an ancient mechanism of gene expression control by RBMX family proteins that predates the radiation of mammals, and provides a new understanding of how ultra-long exons are properly incorporated into mRNAs.

## Results

### RBMX primarily operates as a splicing repressor in somatic cells

We first set out to identify the spectrum of splicing events that are strongly controlled by RBMX across different human cell lines. We used RNA-seq from biological triplicate MDA-MB-231 cells treated with siRNA against RBMX (achieving >90% depletion, *Figure 1B*), followed by bioinformatics analysis using the SUPPA2 (*Trincado et al., 2018*) and MAJIQ (*Vaquero-Garcia et al., 2023*; *Vaquero-Garcia et al., 2016*) splicing prediction tools. We identified 315 changes in RNA processing patterns

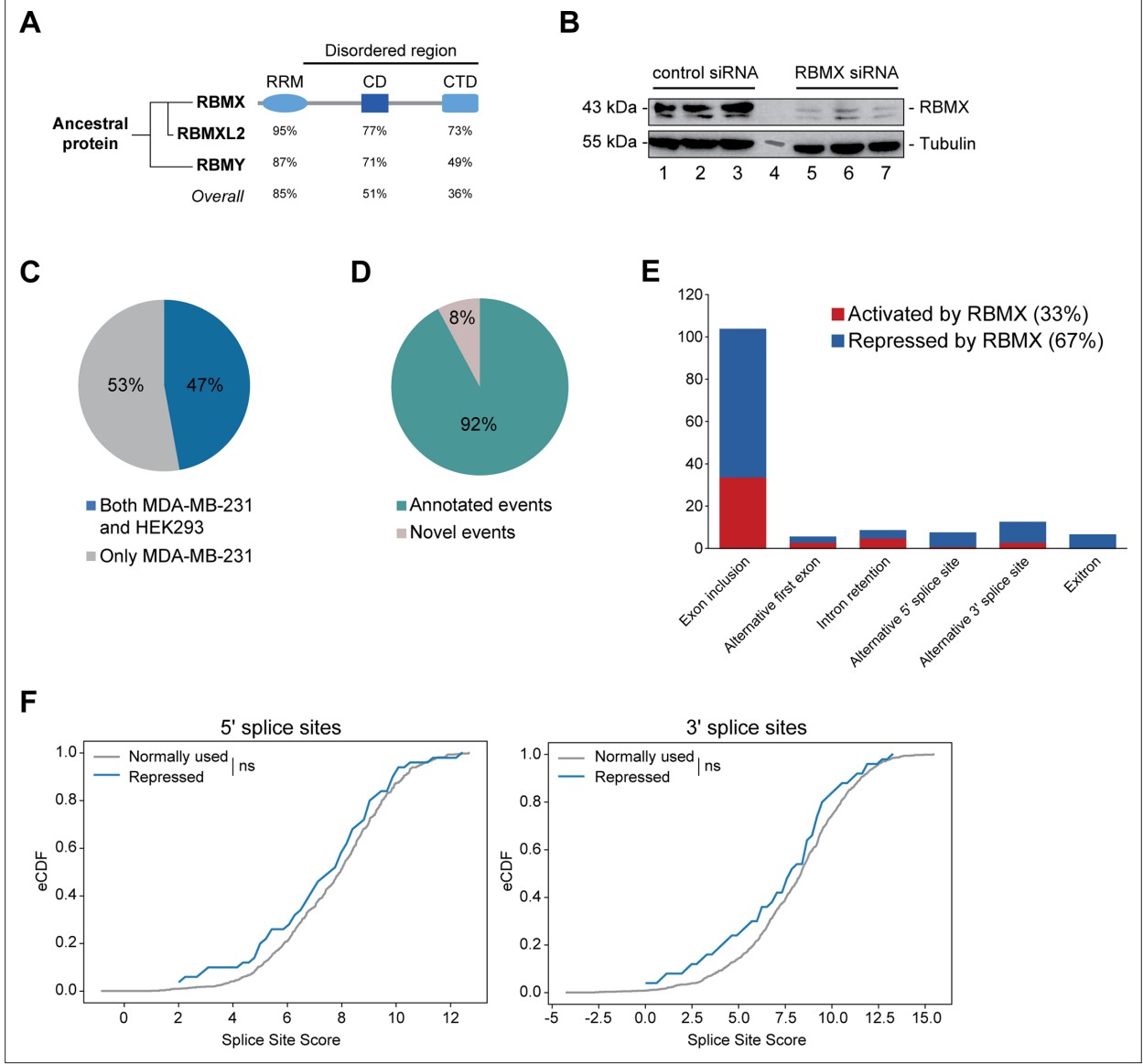

**Figure 1.** RBMX primarily operates as a splicing repressor in human somatic cells. (**A**) Schematic structure of RBMX family proteins (left side, cladogram) and amino acid similarity of each domain between RBMX protein and two other members of this family, RBMXL2 and RBMY. RRM, RNA recognition motif; CD, central domain important for recognition of nascent transcripts and nuclear localisation; CTD, C-terminal domain, involved in RNA binding (*Elliott et al., 2019*). (**B**) Western blot analysis shows efficient siRNA-mediated depletion of RBMX from MDA-MB-231 cells (each lane contains biologically independent replicate, apart from lane 4 which contained size markers). (**C**) Pie chart showing the percentages of events controlled by RBMX in both MDA-MB-231 (this study) and HEK293 (*Liu et al., 2017*) cells. (**D**) Pie chart showing the percentages of events controlled by RBMX in both MDA-MB-231 and HEK293 cells that have been previously annotated (Refseq, Ensembl, Gencode), and those that are novel to this study. (**E**) Bar chart showing the different types of alternative splicing events controlled by RBMX protein in both HEK293 and MDA-MB-231 cells, summarising the proportion of splicing events that are activated by RBMX versus those that are repressed. (**F**) Splice site score analyses for 5′ (left panel) and 3′ (right panel) splice sites repressed by RBMX compared to RBMX non-responsive alternative splice sites. eCDF, empirical Cumulative Distribution Function. Two-sample KS test two-sided p-value = 0.41 and 0.33, respectively.

The online version of this article includes the following source data and figure supplement(s) for figure 1:

**Source data 1.** List of splicing defects in MDA-MB-231 and HEK293 related to *Figure 1C* and *Figure 1—figure supplement 1A*.

**Figure supplement 1.** Splicing patterns controlled by RBMX within MDA-MB-231 cells and splice site strengths of exons activated by RBMX.

in response to RBMX-depletion that were of high enough amplitude to be visually confirmed on the IGV genome browser (*Robinson et al., 2011*; *Figure 1—figure supplement 1A*; *Figure 1—source data 1*). Analysis of these splicing events within existing RNA-seq data from HEK293 cells depleted for RBMX (GSE74085) (*Liu et al., 2017*) revealed 148 high amplitude events that are controlled by RBMX in both HEK293 and MDA-MB-231 cells (*Figure 1C*). We concentrated our downstream analysis on these splicing events (*Figure 1—source data 1*). 92% of the splicing events regulated by RBMX in human somatic cells were already annotated on Ensembl, Gencode or Refseq (*Figure 1D*). Strikingly two-thirds of these events are repressed by RBMX, meaning they were increasingly used in RBMX-depleted cells compared to control, and include exon inclusion, alternative 5′ and 3′ splice sites, exitrons, and intron retention (*Figure 1E*). Furthermore, analysis of splice site strength revealed that, unlike splice sites activated by RBMX (*Figure 1—figure supplement 1B*), alternative splice sites repressed by RBMX have comparable strength to more commonly used splice sites (*Figure 1F*). This means that RBMX operates as a splicing repressor in human somatic cells to prevent the use of 'decoy' splice sites that could disrupt normal patterns of gene expression.

## Splicing control and sites of RBMX protein-RNA interaction are enriched within long internal exons

The above data indicated that RBMX has a major role in repressing cryptic splicing patterns in human somatic cells. To further correlate splicing regulation to patterns of RBMX protein-RNA interactions, we next mapped the distribution of RBMX-RNA binding sites in human somatic cells. We engineered a stable human HEK293 cell line to express RBMX-FLAG fusion protein in response to tetracycline addition. Western blotting showed that expression of RBMX-FLAG was efficiently induced after tetracycline treatment. Importantly, levels of the induced RBMX-FLAG protein were similar to those of endogenous RBMX (*Figure 2A*). We next used this inducible cell line to carry out individual nucleotide resolution crosslinking and immunoprecipitation (iCLIP) – a technique that produces a global picture of protein-RNA binding sites (*Konig et al., 2011*). After crosslinking, RBMX-FLAG protein was immunoprecipitated, then infra-red labeled RNA-protein adducts were isolated (*Figure 2B*) and subjected to library preparation. Following deep sequencing of biological triplicate experiments, 5–10 million unique reads (referred to here as iCLIP tags, representing sites of RBMX protein-RNA cross-linking) were aligned to the human genome. Each individual iCLIP replicate showed at least 70% correlation with each of the others (*Figure 2—figure supplement 1A*). K-mer motif analysis revealed RBMX preferentially binds to AG-rich sequences (*Figure 2C* and *Figure 2—figure supplement 1B*).

In line with previous work on other RNA binding proteins (*Van Nostrand et al., 2020*), only 31% of the RNA splicing events that are controlled by RBMX in both HEK293 cells and MDA-MB-231 cells were identified by iCLIP as direct targets for RBMX binding (*Figure 2—figure supplement 1C*, and *Figure 2—source data 1*). Furthermore, when we plotted the fraction of RBMX iCLIP tags present near exons that contain splicing defects in the absence of RBMX, and compared it to iCLIP tags present near a set of exons unaffected by RBMX depletion, we did not detect significant enrichment of RBMX binding within exons that contain splice sites repressed by RBMX (*Figure 2—figure supplement 1D, E*). There was significant enrichment of RBMX binding seen in flanking introns (downstream for RBMX-repressed and upstream for RBMX-activated). However, RBMX-responsive internal exons that did contain RBMX iCLIP tags were significantly longer than the ones that are not bound by RBMX (*Figure 2D* and *Figure 2—source data 1*). We therefore compared the length of the internal exons regulated (identified by RNA-seq) and bound by RBMX (identified by iCLIP) within protein-coding genes to all internal mRNA exons expressed in HEK293 (*Liu et al., 2017*). We reasoned that larger exons might have a higher chance to be bound by RBMX merely because of their large size. To minimise this effect, we did not take into account the density of RBMX binding and instead considered all exons that contained at least one iCLIP tag. Strikingly, we found that exons regulated and bound by RBMX were significantly longer than the median size of HEK293 mRNA exons which is ~130 bp (*Figure 2E*, and *Figure 2—source data 2*). This led us to test whether RBMX protein is preferentially associated with long exons. For this, we plotted the distribution of internal exons bound and regulated by RBMX together with all internal exons expressed from HEK293 mRNA genes (*Liu et al., 2017*). We found that RBMX controls and binds two different classes of exons: the first has a comparable length to the average HEK293 exon, while the second is extremely long, exceeding 1000 bp in length (*Figure 2F*). We defined this second class as 'ultra-long exons,' which represented the 17.6% of internal exons regulated by RBMX

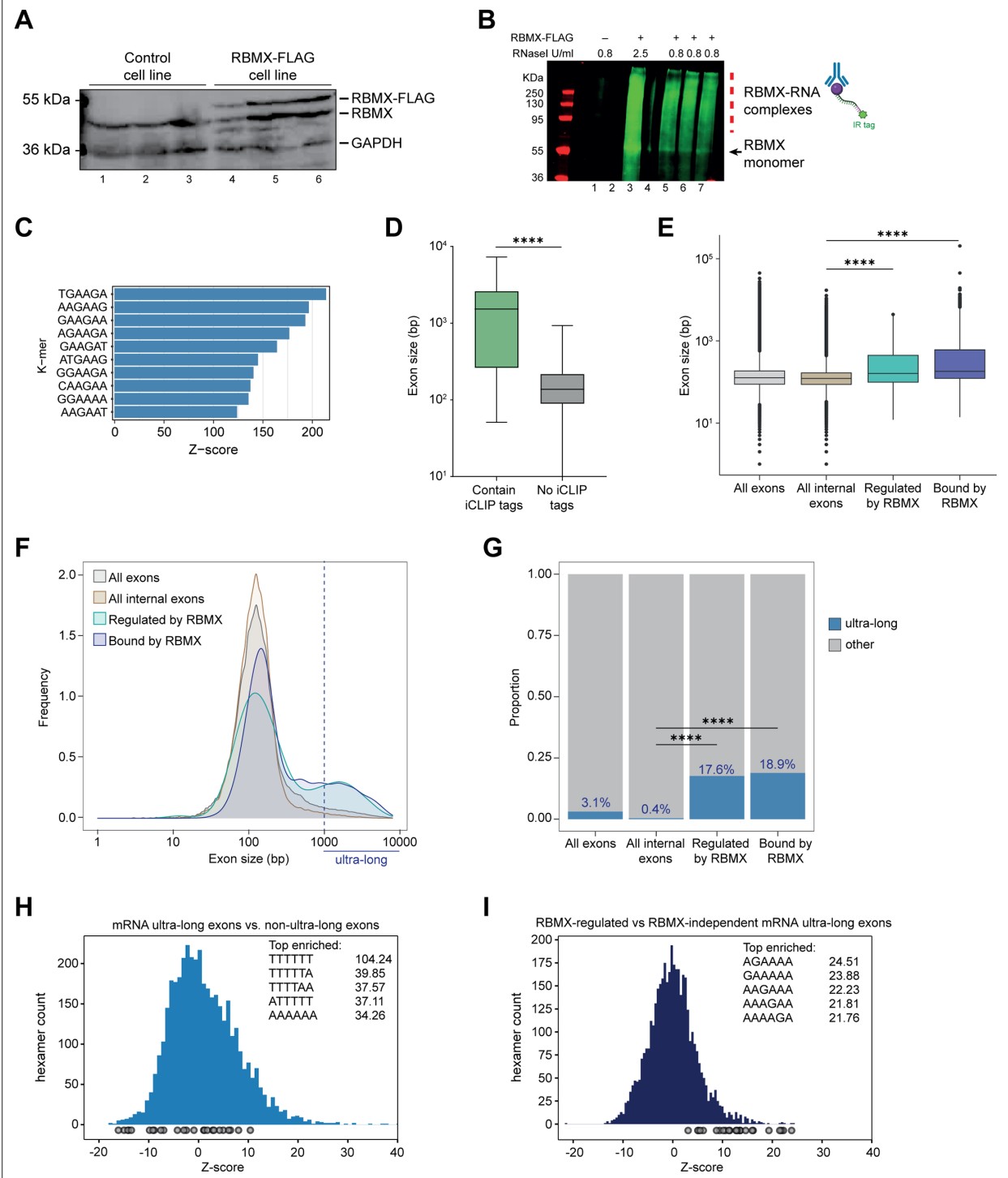

**Figure 2.** Splicing control and sites of RBMX protein-RNA interaction are enriched within long internal exons. (**A**) Western blot showing levels of RBMX-FLAG protein, expressed after 24 hr treatment with tetracycline, compared to endogenous RBMX within HEK293 cells, both detected using α-RBMX antibody. α-GAPDH antibody was used as a loading control. Each lane corresponds to a biologically independent replicate sample. (**B**) RNAs cross-linked to RBMX-FLAG during individual nucleotide resolution crosslinking and immunoprecipitation (iCLIP) were detected through the infrared adaptor (RBMX-RNA complexes). Lane 1, anti-FLAG pull-down from crosslinked HEK293 control cells not expressing RBMX-FLAG proteins, treated with 0.8 U/ml RNaseI. Lane 3, RBMX-FLAG pull-down crosslinked to RNA, treated with 2.5 U/ml RNaseI. Lanes 5–7, RBMX-FLAG pull-down crosslinked to RNA, treated with 0.8 U/ml RNaseI. Samples in lanes 5–7 were used for iCLIP library preparation. Lanes 2 and 4 are empty. (**C**) K-mer analysis shows the top 10 enriched motifs within sequences surrounding RBMX iCLIP tags. (**D**) Boxplot analysis shows sizes of exons containing splicing events regulated by RBMX, grouped by whether they contain CLIP tags or not. ****, p<0.0001 (Mann-Whitney test). (**E**) Boxplot analysis shows the distribution of exon sizes

*Figure 2 continued on next page*

*Figure 2 continued*

relative to: all or internal exons contained in mRNA genes expressed in HEK293 cells (*Liu et al., 2017*); exons regulated by RBMX as identified by RNA-seq; exons containing RBMX binding sites as identified by iCLIP, listed independently of iCLIP tag density. Median sizes for each group are shown. ****, p<0.0001 (Wilcoxon rank test and Kruskal-Wallis test). (F) Distribution plot of exon sizes for the groups shown in (E). Note the increased accumulation of exons larger than 1000 bp (ultra-long exons) in RBMX-bound and regulated exons compared to all exons expressed in HEK293 (*Liu et al., 2017*). (G) Bar plot indicating the proportion of ultra-long exons in the groups shown in (E, F). ****, p<0.0001 (Chi-squared test). (H) Histogram of hexamer Z-scores for ultra-long exons (exceeding 1000 nt) versus non-ultra-long exons from Ensembl canonical mRNA transcripts. The top five enriched hexamers are shown with corresponding Z-scores. Grey dots indicate histogram bins containing one of the top 25 RBMX iCLIP hexamer motifs. (I) Similar analysis as in (H), but for ultra-long exons with evidence of RBMX binding or regulation versus RBMX-independent ultra-long exons.

The online version of this article includes the following source data and figure supplement(s) for figure 2:

**Source data 1.** List of splicing defects with nearby RBMX CLIP tags from HEK293 cells related to *Figure 2D* and *Figure 2—figure supplement 1C*.

**Source data 2.** List of exons analysed in *Figure 2E–G*.

**Figure supplement 1.** Further analysis of RBMX protein-RNA interactions.

and 18.9% of the ones that contained RBMX iCLIP tags. These proportions were significantly enriched compared to the general abundance of internal ultra-long exons expressed from HEK293 cells, which was only 0.4% (*Figure 2G*). K-mer analyses also showed that while ultra-long exons within mRNAs are rich in AT-rich sequences compared to shorter exons (*Figure 2H*), the ultra-long exons that are either regulated or bound by RBMX displayed enrichment of AG-rich sequences (*Figure 2I*), consistent with our identified RBMX-recognised sequences (*Figure 2C*). Overall, this data revealed a function for RBMX in the regulation of splicing of a particular group of ultra-long exons.

## RBMX is important for proper splicing inclusion of full-length ultra-long exons within genes involved in processes including DNA repair and chromosome biology

We next wondered whether ultra-long exons regulated by RBMX (which represented 11.6% of all ultra-long internal exons from genes expressed in HEK293) had any particular feature compared to ultra-long exons that were RBMX-independent. To determine whether RBMX regulates particular classes of genes we performed Gene ontology analysis. Both the genes bound by RBMX (detected using iCLIP, *Figure 3—figure supplement 1A* and *Figure 3—source data 1*) and regulated by RBMX in both MDA-MB-231 and HEK293 cell lines (detected using RNA-seq, *Figure 3—figure supplement 1B* and *Figure 3—source data 1*) each showed individual global enrichment in functions connected to genome stability and gene expression. Similarly, Gene ontology analyses for genes that contained ultra-long exons bound by and dependent on RBMX for correct splicing were enriched in pathways involving cell cycle, DNA repair, and chromosome regulation, compared to all expressed genes with ultra-long exons (*Figure 3A* and *Figure 3—source data 1*). These data are consistent with published observations (*Adamson et al., 2012*; *Munschauer et al., 2018*; *Zheng et al., 2020*) that depletion of RBMX reduces genome stability. In addition, comet assays also detected increased levels of genome instability after RBMX depletion (*Figure 3—figure supplement 1C, D*).

The above data indicated that RBMX-RNA binding interactions and splicing control by RBMX are particularly associated with long internal exons and enriched within classes of genes involved in genome stability. These exons included the 2.1 Kb exon 5 of the *ETAA1* (*Ewings Tumour Associated Antigen 1*) gene, where RBMX potently represses a cryptic 3′ splice site that reduces the size of this exon from 2.1 Kb to 100 bp (*Figure 3B* and *Figure 3—figure supplement 2A*). RT-PCR analysis confirmed that RBMX depletion causes a much shorter version of *ETAA1* exon 5 to prevail, particularly in MDA-MB-231 and NCI-H520 cells, but less in MCF7 cells (*Figure 3C*). *ETAA1* encodes a replication stress protein that accumulates at sites of DNA damage and is a component of the ATR signalling response (*Bass et al., 2016*). Selection of RBMX-repressed cryptic 3′ splice sites within *ETAA1* exon 5 removes a long portion of the open reading frame (*Figure 3—figure supplement 2B*). Consistent with the penetrance of this *ETAA1* splicing defect being sufficiently high to affect protein production, no ETAA1 protein was detectable 72 hr after RBMX depletion from MDA-MB-231 cells (*Figure 3D*).

Another ultra-long exon is found within the *REV3L* gene that encodes the catalytic subunit of DNA polymerase ζ that functions in translesion DNA synthesis (*Martin and Wood, 2019*). RBMX similarly represses a cryptic 3′ splice site within the ultra-long exon 13 of the *REV3L* gene (~4.2 Kb), that has

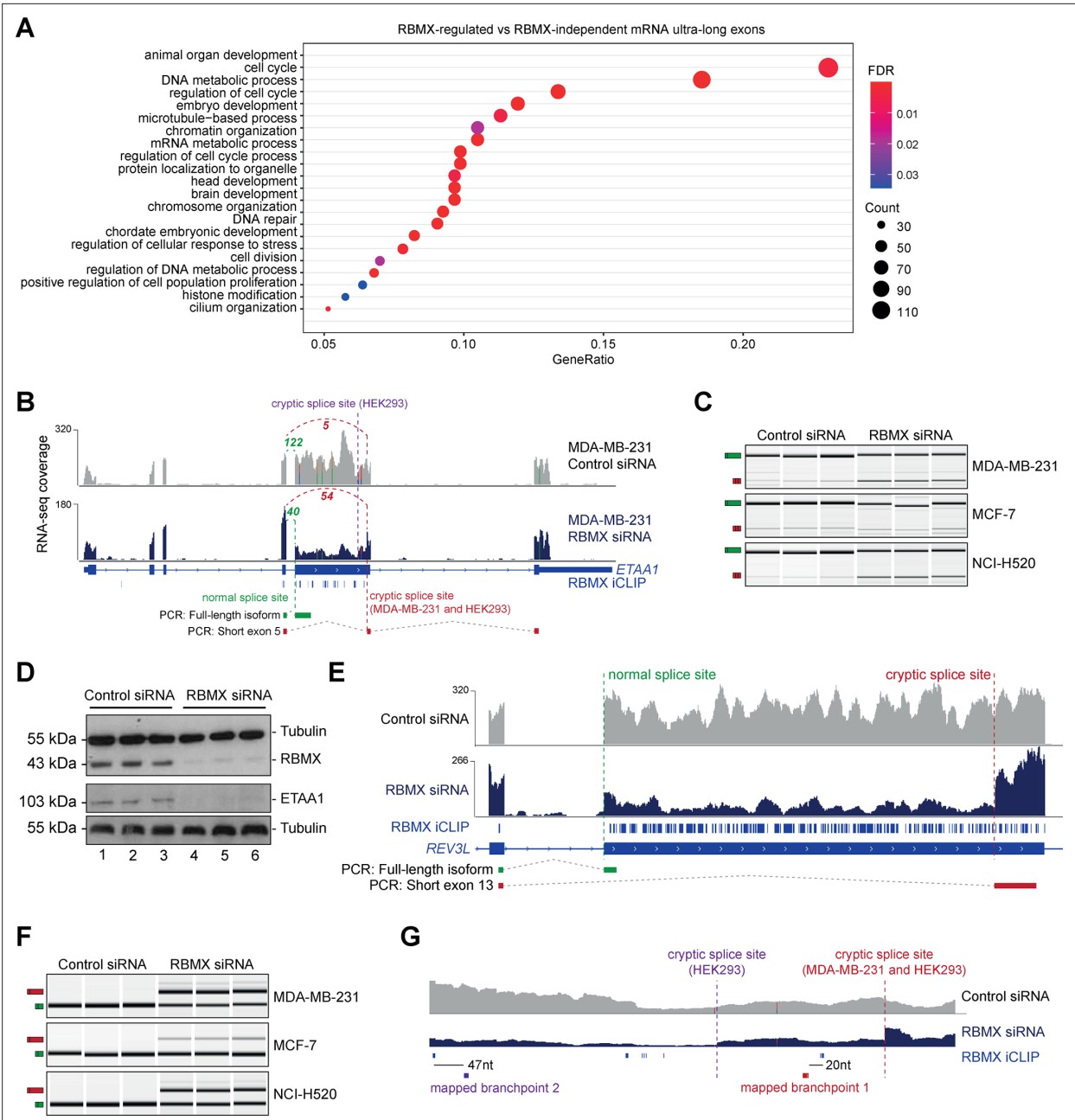

**Figure 3.** RBMX protein is important for full-length splicing inclusion of ultra-long exons involved in DNA repair and chromosome biology. (**A**) Gene ontology analysis of genes with ultra-long exons regulated and bound by RBMX displaying significant gene ontology biological process (GOBP) terms containing at least 5% of the total gene list. FDR, False Discovery Rate. Count, number of genes in the GOBP group. GeneRatio, proportion of genes in the GOBP group relative to the full list of RBMX-regulated genes. (**B**) Snapshot of RNA-seq merged tracks from MDA-MB-231 cells and RBMX iCLIP tags from HEK293 cells from the IGV genome browser shows cryptic 3′ splice sites repressed by RBMX in *ETAA1* exon 5. At the bottom, the schematic of PCR products identified by RT-PCR in (**C**). (**C**) RT-PCR analysis shows splicing inclusion of *ETAA1* exon 5 upon siRNA-mediated depletion of RBMX in the indicated cell lines (separate lanes correspond to analysis of independent biological replicate samples). (**D**) Western blot analysis shows that ETAA1 protein expression is dependent on RBMX. Anti-Tubulin detection was used as loading control (separate lanes correspond to independent biological replicate samples). (**E**) Snapshot of RNA-seq merged tracks from MDA-MB-231 cells and RBMX iCLIP tags from HEK293 cells from the IGV genome browser shows RBMX represses a cryptic 3′ splice site within the ultra-long exon 13 of *REV3L*. At the bottom, the schematic of PCR products identified by RT-PCR in (**F**). (**F**) RT-PCR analysis shows splicing inclusion of *REV3L* exon 13 upon siRNA-mediated depletion of RBMX in the indicated cell lines (separate lanes correspond to the analysis of independent biological replicate samples). (**G**) Snapshot of RNA-seq merged tracks from MDA-MB-231 cells and RBMX iCLIP tags from HEK293 cells from IGV genome browser. The location of experimentally mapped branchpoints relative to RBMX binding is indicated.

*Figure 3 continued on next page*

*Figure 3 continued*

The online version of this article includes the following source data and figure supplement(s) for figure 3:

**Source data 1.** Gene ontology analyses related to *Figure 3A* and *Figure 3—figure supplement 1A, B*.

**Figure supplement 1.** Further analysis of gene categories encoding mRNAs bound by RBMX and association with genome stability.

**Figure supplement 2.** Further analysis of cryptic splicing patterns within the *ETAA1* and *ATRX* mRNAs.

an extremely high density of RBMX binding (*Figure 3E*). RT-PCR analysis confirmed a strong splicing switch to a cryptic splice site within *REV3L* exon 13 after RBMX was depleted from MDA-MB-231, MCF7, and NCI-H520 cells (*Figure 3F*).

We also detected extremely high-density RBMX protein binding within exon 9 of the *ATRX* gene (3 Kb in length) that encodes a chromatin remodelling protein involved in mitosis. Depletion of RBMX results in the expression of a shortened version of *ATRX* exon 9, caused by the formation of an exitron through the selection of cryptic 5′ and 3′ splice sites within exon 9 (*Figure 3—figure supplement 2C*).

## RBMX protein-RNA interactions may insulate important splicing signals from the spliceosome

The iCLIP data suggested a model where RBMX protein binding may insulate ultra-long exons so that cryptic splice sites cannot be accessed by the spliceosome. This model predicted that RBMX binding sites would be close to important sequences used for the selection of cryptic splice sites. RBMX iCLIP tags mapped just upstream of the cryptic 3′ splice sites within *ETAA1* exon 5 in HEK293 cells and MDA-MB-231 cells after RBMX depletion (*Figure 3B*), suggesting that RBMX may bind close to the branchpoints used to generate these cryptic splicing patterns. However, although usually located close to their associated 3′ splice sites, in some cases branchpoints can be located far upstream (*Gooding et al., 2006*). We tested the prediction that RBMX may sterically interfere with components of the spliceosome by directly mapping the branchpoints associated with the use of these cryptic *ETAA1* splice sites. To facilitate mapping of the branchpoint sequences used by the cryptic 3′ splice site within *ETAA1* exon 5, we made a minigene by cloning the ultra-long *ETAA1* exon 5 and flanking intron sequences between constitutively spliced β-globin exons (*Figure 3—figure supplement 2D*). Confirming that this minigene recapitulated cryptic splicing patterns, after transfection into HEK293 cells we could detect splicing inclusion of both the full-length and shorter (cryptic) versions of *ETAA1* exon 5 mRNA isoforms using multiplex RT-PCR (*Figure 3—figure supplement 2E*). We then used an RT-PCR assay (*Figure 3—figure supplement 2F*) to monitor the position of branchpoint just upstream of the cryptic 3′ splice sites of *ETAA1* exon 5 (*Královičová et al., 2021*). Sanger sequencing of the amplification product made in this assay confirmed that the branchpoint sequences used by these cryptic 3′ splice sites are adjacent to RBMX binding sites (*Figure 3G* and *Figure 3—figure supplement 2G*).

## RBMXL2 and RBMY can replace the activity of RBMX in somatic cells

The above data showed that although RBMX can activate the splicing of some exons, it predominantly operates as a splicing repressor in human somatic cells, and moreover has a key role in repressing cryptic splicing within ultra-long exons. This pattern of RBMX activity is thus very similar to that previously reported for RBMXL2 in the germline, where RBMXL2 represses cryptic splice sites during meiosis. RBMXL2 is expressed during male meiosis when the X chromosome is silenced. To directly mimic this switch in protein expression we constructed a HEK293 RBMXL2-FLAG tetracycline-inducible cell line, from which we depleted RBMX using siRNA (*Figure 4A*). Western blots showed that RBMX was successfully depleted after siRNA treatment, and the RBMXL2-FLAG protein was strongly expressed after tetracycline induction, thus simulating their relative expression patterns in meiotic cells (*Figure 4B*). We globally investigated patterns of splicing in these rescue experiments by performing RNA-seq analysis of each of the experimental groups. Strikingly, almost 80% of splicing defects that we could detect after RBMX-depletion were rescued by tetracycline-induced RBMXL2 (*Figure 4C*, and *Figure 4—source data 1*). Exons rescued by RBMXL2 tended to be larger than exons not restored (*Figure 4D*), and most of the splice events that were restored by RBMXL2 expression had nearby RBMX binding sites evidenced by iCLIP (*Figure 4E*). We then validated three cryptic splicing patterns by RT-PCR. Confirming our previous finding, in the absence of tetracycline treatment

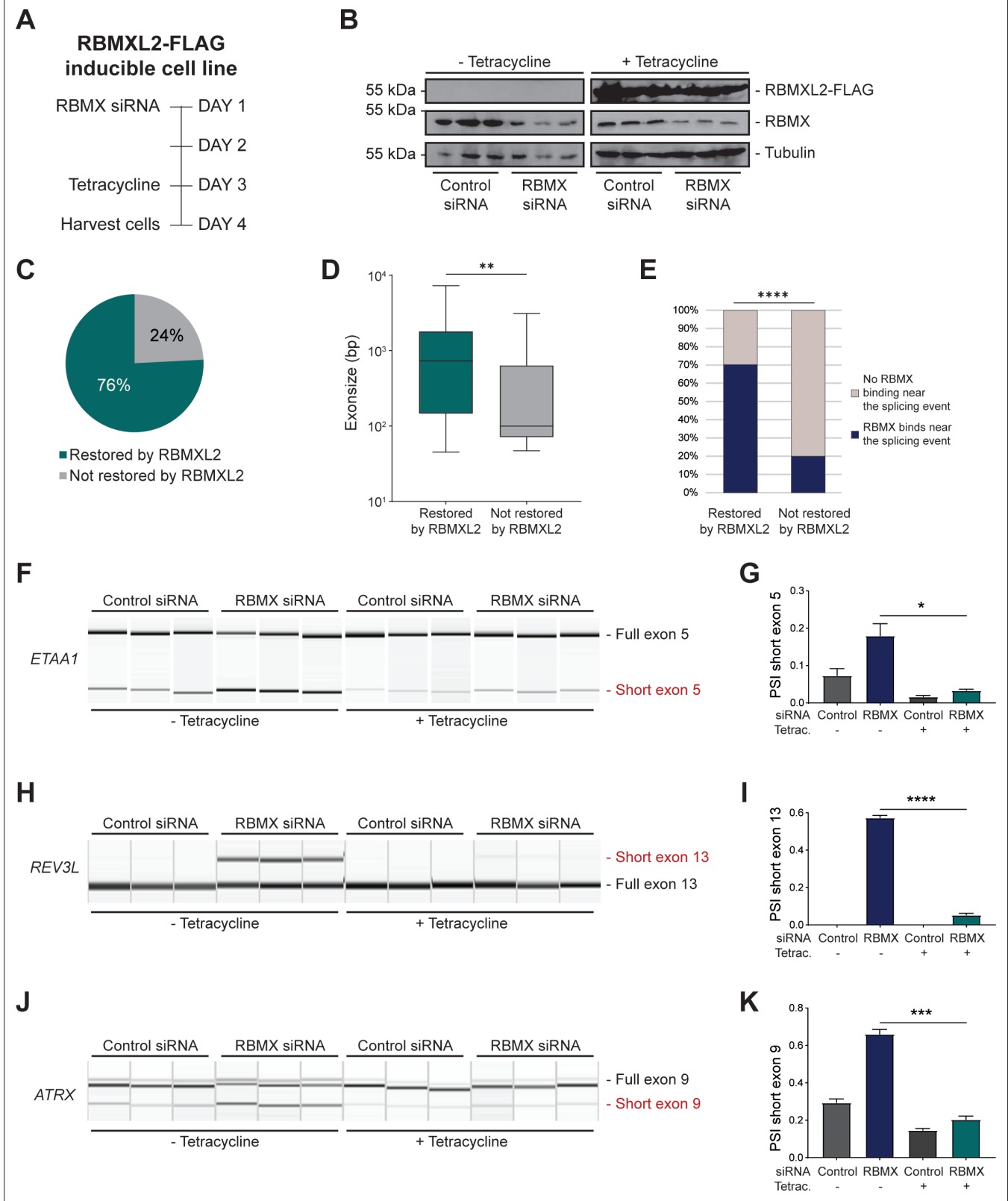

**Figure 4.** RBMXL2 can replace the activity of RBMX in ensuring proper splicing inclusion of ultra-long exons. (**A**) Schematic of the time-course experiment used to analyse human RBMXL2 function in RBMX-depleted HEK293 cells. All conditions were repeated in biological triplicates. (**B**) Western blot analysis shows that RBMXL2-FLAG protein is stably expressed in HEK293 cells after 24 hr of tetracycline induction, and RBMX protein is successfully depleted after 72 hr of siRNA treatment. Separate lanes correspond to independent biological replicate samples. (**C**) Pie chart showing

*Figure 4 continued on next page*

*Figure 4 continued*

the percentage of splicing events detected by RNA-seq that were defective in RBMX-depleted cells and restored by overexpression of RBMXL2. (**D**) Boxplot analysis shows the distribution of exon sizes relative to exons undergoing defective splicing in RBMX-depleted cells, grouped by whether splicing patters were restored by RBMXL2 overexpression. **p<0.01 (Mann-Whitney test). (**E**) Bar plot analysis shows the proportion of exons containing RBMX individual nucleotide resolution crosslinking and immunoprecipitation (iCLIP) tags, grouped by whether splicing patterns were restored by RBMXL2 overexpression. ****p<0.0001 (Chi-squared test). (**F, H, J**) Capillary gel electrophoretograms show RNA processing patterns of endogenous ultra-long exons within *ETAA1*, *REV3L,* and *ATRX* controlled by RBMX and RBMXL2 analysed using isoform-specific RT-PCR. Separate lanes correspond to the analysis of independent biological replicate samples. (**G, I, K**) Bar charts showing percentage splicing inclusion (PSI) of cryptic isoforms from the endogenous *ETAA1*, *REV3L,* and *ATRX* genes under the different experimental conditions, measured in experiments in (**F**), (**H**) and (**J**), respectively. p-values were calculated using an unpaired t-test. * p< 0.05; ***p< 0.001; ****p<0.0001, using n=3 biological replicates for each condition as shown in capillary gel electrophoretograms.

The online version of this article includes the following source data and figure supplement(s) for figure 4:

**Source data 1.** List of splicing defects restored by overexpression of RBMXL2 related to *Figure 4C*.

**Figure supplement 1.** RBMXL2 can replace the activity of RBMX in ensuring proper splicing inclusion of ultra-long exons.

**Figure supplement 2.** RBMY can replace the activity of RBMX in ensuring proper splicing inclusion of ultra-long exons.

depletion of RBMX led to an increased selection of cryptic splice sites within *ETAA1* exon 5 and *REV3L* exon 13, and to the formation of an exitron within *ATRX* exon 9 (*Figure 4F–K*, compare lanes 1–3 with lanes 4–6). Consistent with our RNA-seq analysis (*Figure 4—figure supplement 1A–C*), tetracycline-induction of RBMXL2 was sufficient to repress production of each of these aberrant splice isoforms (*Figure 4F–K*, compare lanes 7–9 with lanes 10–12). These experiments indicate that RBMXL2 is able to replace RBMX activity in regulating ultra-long exons within somatic cells.

RBMX and RBMXL2 are both more distantly related to the Y chromosome-encoded RBMY protein, with RBMX and RBMY diverging when the mammalian Y chromosome evolved (*Figure 1A*). RBMY has also been implicated in splicing control (*Nasim et al., 2003*; *Venables et al., 2000*), but its functions are very poorly understood. We thus tested whether RBMY might also be performing a similar function to RBMX. Employing a HEK293 cell line containing tetracycline-inducible, FLAG-tagged RBMY protein, we detected successful recovery of normal splicing patterns of the ultra-long exons within the *ETAA1*, *REV3L,* and *ATRX* genes within RBMX-depleted cells 24 hr after tetracycline induction of RBMY (*Figure 4—figure supplement 2*). These results indicate that even despite its more extensive divergence, RBMY can also functionally replace RBMX in cryptic splice site control within long exons. Thus, splicing control mechanisms by RBMX family proteins pre-date the evolution of the mammalian X and Y chromosomes.

## The disordered domain of RBMXL2 is required for efficient splicing control of ultra-long exons

The above data showed that RBMX predominantly operates as a splicing repressor in somatic cells, thus performing a functionally parallel role to RBMXL2 in the germline. Although RBMX contains an RRM domain that is the most highly conserved region compared with RBMXL2 and RBMY, splicing activation by RBMX depends on its C-terminal disordered domain that also binds to RNA (*Liu et al., 2017*; *Moursy et al., 2014*). We thus reasoned that if RBMX and RBMXL2 were performing equivalent molecular functions, the rescue of splicing by RBMXL2 should be mediated by the disordered region of RBMXL2 alone, independent of the RRM (*Liu et al., 2017*; *Moursy et al., 2014*). To test this prediction, we created a new tetracycline-inducible HEK293 cell line expressing the disordered region of the RBMXL2 protein and not the RRM domain (RBMXL2ΔRRM, *Figure 5A*). Tetracycline induction of this RBMXL2ΔRRM protein was able to rescue siRNA-mediated depletion of RBMX (*Figure 5B–G*), directly confirming that the C-terminal disordered domain of RBMXL2 protein is responsible for mediating cryptic splicing repression.

## Discussion

We previously showed that the germ cell-specific RBMXL2 protein represses cryptic splice site selection during meiotic prophase. Here, we find that this is part of a bigger picture, where the closely related but more ubiquitously expressed RBMX protein also provides a similar activity within somatic cells. Supporting this conclusion, both RBMX and RBMXL2 proteins most frequently operate as

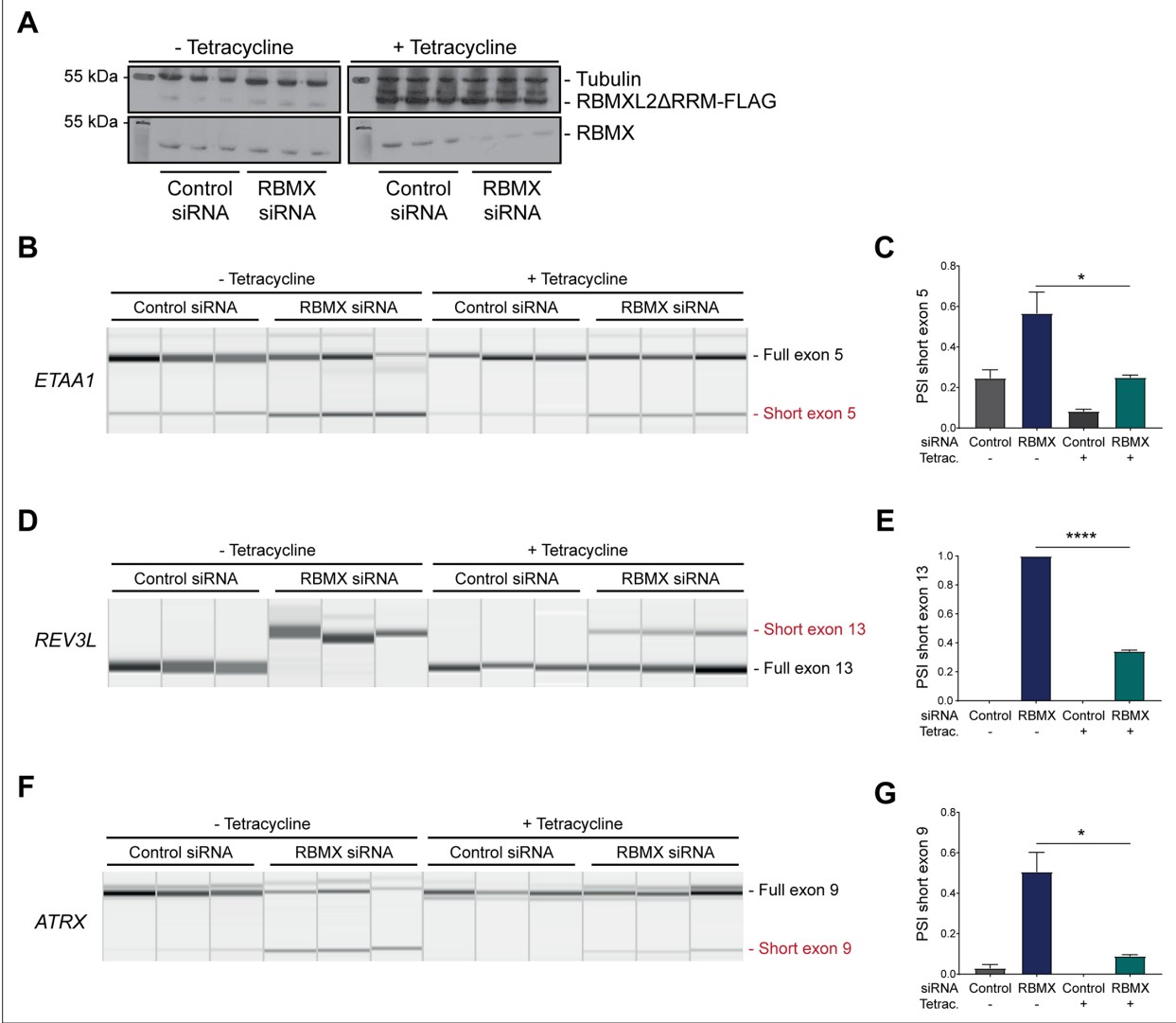

**Figure 5.** The disordered domain of RBMXL2 is required to mediate splicing control of ultra-long exons in HEK293 cells. (**A**) Western blot analysis shows that RBMXL2ΔRRM-FLAG protein is stably expressed in HEK293 cells after 24 hr of tetracycline induction, and RBMX protein is successfully depleted after 72 hr of siRNA treatment. Separate lanes correspond to the analysis of independent biological replicate samples. (**B, D, F**) Capillary gel electrophoretograms show RNA processing patterns of endogenous ultra-long exons within *ETAA1*, *REV3L*, and *ATRX* analysed using isoform-specific RT-PCR. Separate lanes correspond to the analysis of independent biological replicate samples. (**C, E, G**) Bar charts showing percentage splicing inclusion (PSI) of cryptic isoforms from the endogenous *ETAA1*, *REV3L,* and *ATRX* genes under the different experimental conditions, measured in experiments in (**B**), (**D**), and (**F**), respectively. p-values were calculated using an unpaired t-test. ****p<0.0001, using biologically independent sampleas.

splicing repressors in their respective cell types. We further find that RBMX binds and is key for proper splicing inclusion of a group of ultra-long exons, defined as exceeding 1 Kb in length. RBMXL2 similarly represses cryptic splice sites within ultra-long exons of genes involved in genome stability including *Brca2* and *Meioc* (**Ehrmann et al., 2019**). Furthermore, RBMXL2 and even the more diverged RBMY protein are able to provide a direct replacement for RBMX splicing control within human somatic cells. Although many of the splice sites within ultra-long exons we find to be repressed by RBMX are already annotated, they are not usually selected in the human cell lines we investigated and thus represent potential decoy splice sites that would interfere with full-length gene expression.

Long human exons provide an enigma in understanding gene expression. Most human exons have evolved to be quite short (~130 bp) to facilitate a process called exon definition, in which protein-protein interactions between early spliceosome components bound to closely juxtaposed splice

sites promote full spliceosome assembly (**Black, 1995**; **Robberson et al., 1990**). Exon definition also requires additional RNA binding proteins to recognise exons and flanking intron sequences. These include members of the SR protein family that bind to exonic splicing enhancers (ESEs) and activate exon inclusion, with exons typically having higher ESE content relative to introns. While the mechanisms that ensure proper splicing inclusion of long exons are not well understood, cryptic splice sites would be statistically more likely to occur within long exons compared to short exons, where they could prevent full-length exon inclusion. Cryptic splice sites within long exons could be particularly problematic (compared to an intronic location) since they would be embedded within a high ESE sequence environment. For example, some long exons require interactions with the SR protein SRSF3 and hnRNP K and phase separation of transcription factors to be spliced (**Kawachi et al., 2021**). Hence, although the functions of hnRNPs in repressing cryptic splice events have often concentrated on their role within introns, other hnRNPs as well as RBMX might also show enriched binding within ultra-long exons to help repress cryptic splice site selection.

The X chromosome is required for viability. This means that meiotic sex chromosome inactivation (inactivation of the X and Y chromosomes during meiosis) coordinately represses a panel of essential genes on the X chromosome, thus opening the need for alternative routes to fulfil their function (**Turner, 2015**). A number of essential X-linked genes have generated autosomal retrogenes that are expressed during meiosis, although genetic inactivation of some of these retrogenes causes a phenotype that manifests outside of meiosis (**Wang, 2004**). An exception is exemplified by the RPL10 and RPL10L proteins that are 95% identical: *RPL10* mutation causes meiotic arrest, and RPL10L has been shown to directly replace its X-linked ortholog *RPL10* during meiosis (**Jiang et al., 2017**; **Wang, 2004**). *RBMXL2* is the only other X-linked retrogene that has been shown to be essential for meiotic prophase (**Ehrmann et al., 2019**). Here, we show that ectopic expression of RBMXL2 can compensate for the lack of RBMX in somatic cells. This is consistent with a recent model suggesting that RBMXL2 directly replaces RBMX function during meiosis because of transcriptional inactivation of the X chromosome (**Aldalaqan et al., 2022**). This general requirement for functionally similar RBMX family proteins across somatic and germ cells further suggests that RBMX-family functions in splicing control

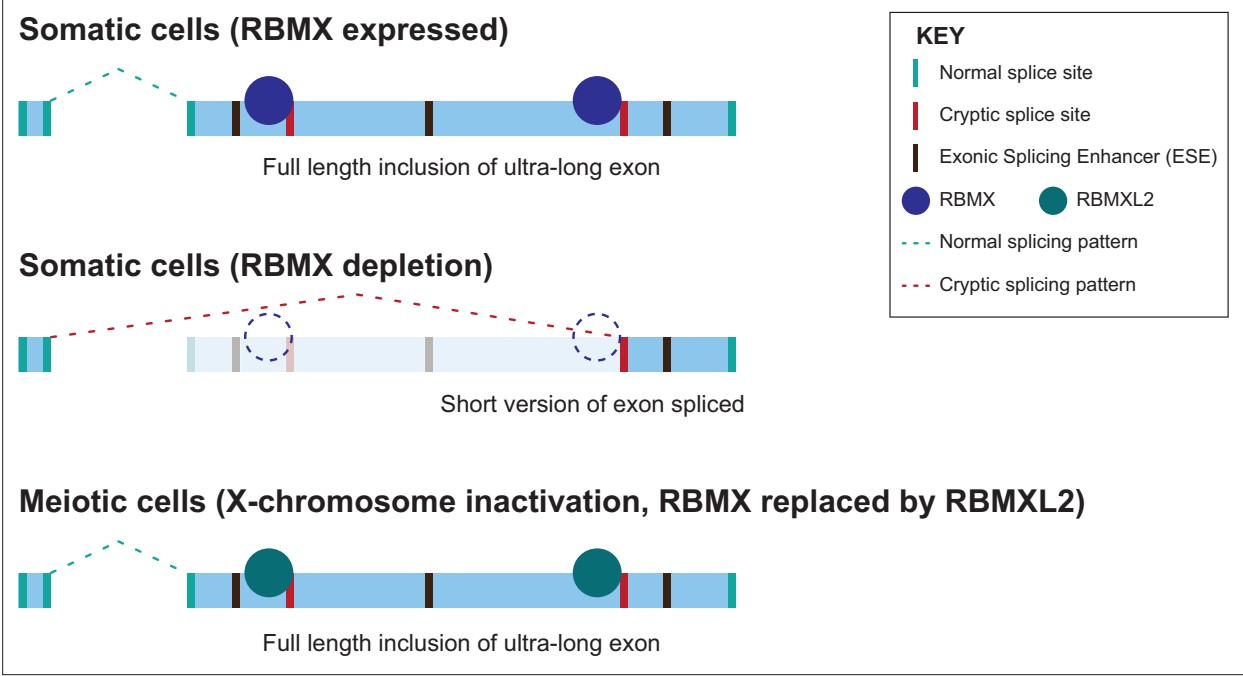

**Figure 6.** Model of cryptic splice site repression within ultra-long exons by RBMX family proteins. Ultra-long exons may be intrinsically fragile as they could contain cryptic splice sites within an environment rich in exonic splicing enhancers (ESEs). RBMX protein binding within ultra-long exons may directly block access of spliceosome components to cryptic splice sites, and depletion of RBMX from somatic cells activates the selection of cryptic splice sites. This means a shorter version of the originally ultra-long exon is included, that fits more easily with exon definition rules normally followed for median-size exons. During meiosis, the lack of RBMX caused by X chromosome inactivation is compensated by the expression of RBMXL2 protein.

have been required for ~200 million years, since before the divergence of separate *RBMX* and *RBMY* genes early in mammalian evolution.

The iCLIP data reported here show a high density of RBMX binding within ultra-long exons, consistent with a model in which RBMX protein binds to RNA mask sequences required for cryptic splice site selection. Such RBMX binding would block access to spliceosome components or splicing activator proteins (*Figure 6*). Our data show that the C-terminal disordered domain of RBMXL2 protein is sufficient to control splicing inclusion of ultra-long exons. This is exactly analogous to the mechanism of control of splicing activation by RBMX, which occurs via recognition of m6A-modified RNA targets via the C-terminal disordered domain (*Liu et al., 2017*). Intriguingly, global studies have shown that m6A residues are enriched within some long internal exons (*Dominissini et al., 2012*), where they might help facilitate RBMX protein-RNA interactions. The C-terminal disordered region of RBMX is also reported to mediate protein-protein interactions, therefore, shorter exons that show defective splicing in RBMX-depleted cells but are not directly bound by RBMX could rely on different regulatory mechanisms. RBMY, RBMX and RBMXL2 directly interact with the SR protein Tra2β (*Elliott et al., 2000*; *Venables et al., 2000*) and have opposing functions during RNA binding and splicing regulation (*Nasim et al., 2003*; *Venables et al., 2000*). Hence, it is still possible that RBMX family proteins counteract recognition by SR proteins of ESEs near cryptic splice sites via a protein-protein interaction mechanism.

Extensive literature shows that RBMX is important for genome stability, including being involved in replication fork activity (*Munschauer et al., 2018*; *Zheng et al., 2020*), sensitivity to genotoxic drugs (*Adamson et al., 2012*) and cell proliferation (https://orcs.thebiogrid.org/Gene/27316). Interestingly, many of the ultra-long exons controlled by RBMX are within genes important for genome stability, including *REV3L, ATRX,* and *ETAA1*. This makes it likely that RBMX contributes to maintaining genome stability by ensuring full-length protein expression of genes important in this process. As an example, we show here that depletion of RBMX protein causes aberrant selection of a high amplitude cryptic splice site within *ETAA1* exon 5 which prevents detectable expression of ETAA1 protein, and contributes to genome instability (*Bass et al., 2016*). Cancer and neurological disorders are amongst the most common human diseases associated with defective DNA damage response (*Jackson and Bartek, 2009*). The double role of RBMX in genome maintenance via both direct participation in the DNA damage response and splicing regulation of genome stability genes could explain why mutations of *RBMX* are associated with an intellectual disability syndrome (*Cai et al., 2021*; *Shashi et al., 2015*), and why RBMX has been identified as a potential tumour suppressor (*Adamson et al., 2012*; *Elliott et al., 2019*). The data reported in this paper thus have implications for understanding the links between RNA processing of unusual exons, genome stability, and intellectual disability.

# Materials and methods

**Key resources table**

| Reagent type (species) or resource | Designation | Source or reference | Identifiers | Additional information |
|---|---|---|---|---|
| Gene (*Homo sapiens*) | *RBMX* | GenBank | NCBI Gene: 27316 Ensembl: ENSG00000147274 | |
| Gene (*Homo sapiens*) | *RBMXL2* | GenBank | NCBI Gene: 27288 Ensembl: ENSG00000170748 | |
| Gene (*Homo sapiens*) | *RBMY/ RBMY1A1* | GenBank | NCBI Gene: 5940 Ensembl: ENSG00000234414 | |
| Gene (*Homo sapiens*) | *ETAA1* | GenBank | NCBI Gene: 54465 Ensembl: ENSG00000143971 | |
| Gene (*Homo sapiens*) | *REV3L* | GenBank | NCBI Gene: 5980 Ensembl: ENSG00000009413 | |
| Gene (*Homo sapiens*) | *ATRX* | GenBank | NCBI Gene: 546 Ensembl: ENSG00000085224 | |

*Continued on next page*

*Continued*

| Reagent type (species) or resource | Designation | Source or reference | Identifiers | Additional information |
|---|---|---|---|---|
| Cell line (*Homo sapiens*) | MDA-MB-231 | ATCC | HTB-26 | |
| Cell line (*Homo sapiens*) | MCF7 | ATCC | HTB-22 | |
| Cell line (*Homo sapiens*) | U-2 OS | ATCC | HTB-96 | |
| Cell line (*Homo sapiens*) | NCI-H520 | ATCC | HTB-182 | |
| Cell line (*Homo sapiens*) | Flp-In–293 Flp-In. T-REx.-293 Cell Line | Thermofisher | R78007 | Used to make stable over-expressing cells. |
| Cell line (*Homo sapiens*) | Flp-In–293 *FLAG-RBMX* | This study | | Used to express tagged RBMX protein in human cells. Available from Elliott lab. |
| Cell line (*Homo sapiens*) | Flp-In–293 *FLAG-RBMXL2* | This study | | Used to express tagged RBMXL2 protein in human cells. Available from Elliott lab. |
| Cell line (*Homo sapiens*) | Flp-In–293 *FLAG-RBMY* | This study | | Used to express tagged RBMY protein in human cells. Available from Elliott lab. |
| Cell line (*Homo sapiens*) | Flp-In–293 *FLAG-RBMXL2ΔRRM* | This study | | Used to express tagged RBMX protein without the RRM in human cells. Available from Elliott lab. |
| Chemical compound | Tetracycline | Sigma-Aldrich | T7660 | |
| Recombinant DNA reagent | pOG44 (plasmid) | Thermofisher Scientific | | Expresses Flp recombinase. |
| Recombinant DNA reagent | FLAG-pcDNA5 (plasmid) | Thermofisher Scientific | | Expression of constructs in the stable Cell lines was induced by treatment with 1 µg/ml tetracycline for 24 hr. |
| Recombinant DNA reagent | pXJ41 (miniGene plasmid) | **Bourgeois et al., 1999** | | |
| Sequence-based reagent | siRNAs | IDT | hs.Ri.RBMX.13.1, hs.Ri.RBMX.13.2 | |
| Antibody | anti-FLAG mouse monoclonal antibody | Sigma-Aldrich | Cat# F1804 RRID:AB_262044 | CLIP (5 µg), WB (1:2000) |
| Antibody | Normal mouse IgG (mouse antibody) | Santa Cruz Biotechnology | Cat# sc-2025 RRID:AB_737182 | CLIP: 5 µg |
| Antibody | Anti-RBMX (rabbit polyclonal antibody) | Cell Signalling Technology | Cat# D7C2V RRID:AB_2798614 | WB (1:1000) |
| Antibody | Anti-ETAA1 (rabbit polyclonal antibody) | Sigma-Aldrich | Cat# HPA035048 RRID:AB_10670300 | WB (1:1000) |
| Antibody | Anti-alpha tubulin (rabbit polyclonal antibody) | Abcam | Cat# ab18251 RRID:AB_2210057 | WB (1:2000) |
| Antibody | Anti-GAPDH (mouse monoclonal antibody) | Abcepta | Cat# P04406 | WB (1:2000) |
| Commercial assay or kit | RNeasy Plus Mini Kit | Qiagen | Cat#74134 | |
| Commercial assay or kit | Comet Assay Kit | Abcam | ab238544 | |
| Software, algorithm | Fastqc | https://anaconda.org/ | | See: **Andrews, 2010** |
| Software, algorithm | STAR | https://anaconda.org/ | v.2.4.2 | See: **Dobin et al., 2013** |
| Software, algorithm | Salmon | https://anaconda.org/ | v.0.9.1 | See: **Patro et al., 2017** |

*Continued on next page*

*Continued*

| Reagent type (species) or resource | Designation | Source or reference | Identifiers | Additional information |
|---|---|---|---|---|
| Software, algorithm | Hisat2 | https://anaconda.org/ | v.2.2.1 | See: *Kim et al., 2015* |
| Software, algorithm | Samtools | https://anaconda.org/ | v.1.14 | See: *Li et al., 2009* |
| Software, algorithm | MAJIQ | https://anaconda.org/ | | See *Vaquero-Garcia et al., 2016* |
| Software, algorithm | SUPPA2 | https://anaconda.org/ | | See: *Trincado et al., 2018* |
| Software, algorithm | iCLIPlib | Github, https://github.com/sudlab/iCLIPlib, copy archived at *Sudbery et al., 2022*. | | |
| Software, algorithm | iMAPS | https://imaps.goodwright.com/ | | |
| Software, algorithm | R/RStudio | Cran | v.3.5.1 | |
| Software, algorithm | DESeq2 | https://bioconductor.org/ | v.1.16.1 | See: *Love et al., 2014* |
| Software, algorithm | biomaRt | https://bioconductor.org/ | v.2.52.0 | See: *Durinck et al., 2005* |
| Software, algorithm | clusterProfiler::enrichGO | https://bioconductor.org/ | v.4.4.4 | See: *Yu et al., 2012* |
| Software, algorithm | Revigo | https://bioconductor.org/ | | See: *Supek et al., 2011* |
| Software, algorithm | ggplot2 | https://bioconductor.org/ | v.3.3.6 | See: *Wickham, 2016* |
| Software, algorithm | ChIPseeker | https://bioconductor.org/ | v.1.32.0 | See: *Yu et al., 2015* |
| Software, algorithm | Pseudorank | https://bioconductor.org/ | v.1.0.1 | See: *Happ et al., 2020* |
| Software, algorithm | Integrative Genomics Viewer | https://igv.org/ | | See: *Robinson et al., 2011* |
| Software, algorithm | GraphPad Prism | https://www.graphpad.com | v.9.5.0 | |
| Software, algorithm | MedCalc | https://medcalc.org/ | 20.218 | |

## Cell culture and cell lines

MDA-MB-231 (ATCC HTB-26), MCF7 (ATCC HTB-22), U-2 OS (ATCC HTB-96), and NCI-H520 (ATCC HTB-182) cells were maintained in Dulbecco's Modified Eagle's Medium (DMEM) high glucose pyruvate medium (Gibco, #10569010), supplemented with 10% fetal bovine serum (FBS, Gibco, #21875034), and 1% Penicillin-Streptomycin (Gibco, #15140130). HEK293 (ATCC CRL-1573) were maintained in DMEM plus 10% fetal bovine serum. Cell line validation was carried out using STR profiling according to the ATCC guidelines. All cell lines underwent regular mycoplasma testing.

## Generation of tetracycline-inducible cell lines

*RMBX*, *RBMXL2*, *RBMY,* and *RBMXL2ΔRRM* genes were cloned onto a FLAG-pcDNA5 vector and co-transfected with pOG44 plasmid into Flp-In T-REX HEK293 cells as previously described (*Ehrmann et al., 2016*). RBMX-FLAG, RBMXL2-FLAG, RBMY-FLAG, and RBMXL2ΔRRM-FLAG expression was induced by the addition of 1 µg/ml tetracycline (Sigma-Aldrich) to promote expression via a tetracycline-inducible promoter. The Flp-In HEK293 cells were cultured in high glucose pyruvate medium (Gibco, #10569010), supplemented with 10% FBS (Gibco, #21875034), and 1% Penicillin-Streptomycin (Gibco, #15140130).

## siRNA knockdown and tetracycline induction

RBMX transient knockdown was established using two different pre-designed siRNAs targeting *RBMX* mRNA transcripts (hs.Ri.RBMX.13.1 and hs.Ri.RBMX.13.2, from Integrated DNA Technologies). Negative control cells were transfected with control siRNA (Integrated DNA Technologies, # 51-01-14-04). Cells were seeded onto six-well plates forward transfected with Lipofectamine RNAiMAX transfection reagent (Invitrogen, # 13778150) according to manufacturer's instructions using 30 pmol of siRNA for 72 hr at 37 °C before harvesting. For tetracycline-inducible cell lines, Flp-In HEK293 cells expressing either RBMXL2-FLAG, or RBMY-FLAG, or RBMXL2ΔRRM-FLAG genes were similarly seeded onto six-well plates and treated with RBMX and control siRNAs for 72 hr at 37 °C. 24 hr before harvesting 1 µg/

ml of tetracycline (Sigma-Aldrich) was added to half of the siRNA-treated samples to promote the expression of FLAG-tagged proteins.

## RNA-seq

RNA was extracted from cells using RNeasy Plus Mini Kit (Qiagen #74134) following the manufacturer's instructions and re-suspended in nuclease-free water. RNA samples were DNase treated (Invitrogen, AM1906). For siRNA-treated MDA-MB-231 cells, paired-end sequencing was done initially for two samples, one of negative control and one of RBMX knock-down, using an Illumina NextSeq 500 instrument. Adapters were trimmed using trimmomatic v.0.32. Three additional biological repeats of negative control and RBMX siRNA-treated MDA-MB-231 cells were then sequenced using an Illumina HiSeq 2000 instrument. The base quality of raw sequencing reads was checked with FastQC (*Andrews, 2010*). RNA-seq reads were mapped to the human genome assembly GRCh38/hg38 using STAR v.2.4.2 (*Dobin et al., 2013*) and subsequently quantified with Salmon v.0.9.1 (*Patro et al., 2017*) and DESeq2 v.1.16.1 (*Love et al., 2014*) on R v.3.5.1. All snapshots indicate merged tracks produced using samtools (*Li et al., 2009*) and visualised with IGV (*Robinson et al., 2011*). For HEK293 cells treated with either RBMX or control siRNA, either in the absence or in the presence of tetracycline, RNAs were sequenced using an Illumina NextSeq 500 instrument. Quality of the reads was checked with FastQC (*Andrews, 2010*). Reads were then aligned to the human genome assembly GRCh38/hg38 to produce BAM files using hisat2 v.2.2.1 (*Kim et al., 2015*) and samtools v.1.14 (*Li et al., 2009*) and visualised using IGV (*Robinson et al., 2011*).

## Identification of splicing changes

Initial comparison of single individual RNA-seq samples from RBMX-depleted and control cells was carried out using MAJIQ (*Vaquero-Garcia et al., 2016*), which identified 596 unique local splicing variations (LSV) at a 20% dPSI minimum cut off from 505 different genes potentially regulated by RBMX. These LSVs were then manually inspected using the RNA-seq data from the second RNA sequencing of biological replicates for both RBMX-depleted and control cells, by visual analysis on the UCSC browser (*Karolchik et al., 2014*) to identify consistent splicing changes that depend on RBMX expression. The triplicate RNA-seq samples were further analysed for splicing variations using SUPPA2 (*Trincado et al., 2018*), which identified 6702 differential splicing isoforms with p<0.05. Predicted splicing changes were confirmed by visual inspection of RNA-seq reads using the UCSC (*Karolchik et al., 2014*) and IGV (*Robinson et al., 2011*) genome browsers. Identification of common splicing changes between RBMX-depleted MDA-MB-231 and HEK293 cells was done by comparing data from this study with data from GSE74085 (*Liu et al., 2017*). For comparative analysis, a negative set of cassette exons that were non-responsive to RBMX depletion were those where every splice junction had an absolute dPSI of 2% or less in two of the knockdown experiments analysed.

## iCLIP

iCLIP experiments were performed on triplicate samples in RBMX-FLAG expressing Flp-In HEK293 cells using the protocol described in *Huppertz et al., 2014*. Briefly, cells were grown in 10 cm tissue culture dishes and irradiated with 400 mJ cm−2 ultraviolet-C light on ice, lysed, and sonicated using Diagenode Bioruptor Pico sonicator for 10 cycles with alternating 30 s on/off at low intensity and 1 mg of protein was digested with 4 U of Turbo DNase (Ambion, AM2238) and 0.28 U/ml (low) or 2.5 U/ml (high) of RNAse I (Thermo Scientific, EN0602). The digested lysates were immunoprecipitated with Protein G Dynabeads (Invitrogen, #10003D) and either 5 μg anti-FLAG antibody (Sigma-Aldrich, F1804) or 5 μg IgG (Santa Cruz biotechnology, sc-2025). Subsequently, a pre-adenylated adaptor L3-IR-App (*Zarnegar et al., 2016*) was ligated to the 3' of the RNA fragments. The captured Protein-RNA complexes were visualised using Odyssey LI-COR CLx imager scanning in both the 700 nm and 800 nm channels. The RNA bound to the proteins was purified, and reverse transcribed with barcoded RT oligos complementary to the L3 adaptor. The cDNAs were purified using Agencourt AMPure XP beads (Beckman Coulter, A63880), circularised, and linearised by PCR amplification. The libraries were gel purified and sequenced on Illumina NextSeq 500. All iCLIP sequencing read analysis was performed on the iMaps webserver (imaps.goodwright.com) using standardised icount demultiplex and analyse work flow. Briefly, reads were demultiplexed using the experimental barcodes, UMIs (unique molecular identifiers) were used to remove PCR duplicates, and reads were mapped to the

human genome sequence (version hg38/GRCh37) using STAR (*Dobin et al., 2013*). Crosslinked sites were identified on the iMAPS platform and the iCount group analysis workflow was used to merge the replicate samples. For enrichment analysis of RBMX iCLIP around cassette exons we compared the number of exons that contained iCLIP binding events that were regulated by RBMX (either repressed or activated) versus non-responsive RBMX cassette exons sets (defined above) in each of the following regions: the proximal intronic region within 300 nt upstream of the 3′ splice site, the proximal intronic region within 300 nt downstream of the 5′ splice site, and the splice site proximal exonic regions within 50 nt of the 3′ splice site or the 5′ splice site.

## K-mer enrichment analysis

K-mer motif enrichment was performed with the z-score approach using the kmer_enrichment.py script from the iCLIPlib suite of tools (https://github.com/sudlab/iCLIPlib, copy archived at *Sudbery et al., 2022*). All transcripts for each non-overlapping protein-coding gene from the Ensembl v.105 annotations were merged into a single transcript, used for this analysis, using cgat gtf2gtf `--method=merge-transcripts` (*Sims et al., 2014*). Each crosslinked base from the merged replicate bam file was extended 15 nucleotides in each direction. For every hexamer, the number of times a crosslink site overlaps a hexamer start position was counted within the gene and then summed across all genes. This occurrence was also calculated across 100 randomisations of the crosslink positions within genes. The z-score was thus calculated for each hexamer as (occurrence – occurrence in randomised sequences)/standard deviation of occurrence in randomised sequences. For motif enrichment analysis within ultra-long internal exons, we compared hexamer occurrence within the set of internal exons from Ensembl v.105 mRNA canonical transcripts of 1000 nt or more and compared those to internal exons of less than 1000 nt and calculated a z-score for each hexamer. A similar analysis was done by stratifying the set of ultra-long internal exons to those with RBMX binding or splicing regulation compared to those with no evidence of RBMX activity.

## Exon size analysis

Analyses of exon sizes from RNA-seq data (*Figures 2D and 4D*) were used using GraphPad Prism 9.5.0. Annotations of all human exons related to position and size were downloaded from Ensembl Genes v.105 (http://www.ensembl.org/biomart/). Selection of exons expressed in HEK293 was performed using data from control RNA-seq samples of the dataset GSE74085 (*Liu et al., 2017*), subsequently filtered to focus on mRNA exons using biomaRt v.2.52.0 (*Durinck et al., 2005*). Size of the internal mRNA exons containing RBMX-regulated splicing patterns was annotated using IGV (*Robinson et al., 2011*). iCLIP tags were extended to 80 nt sequences centered at the crosslinked site, and annotated within human exons using ChIPseeker v.1.32.0 (*Yu et al., 2015*) and Ensembl Genes v.105. iCLIP tags present in mRNA exons were filtered using biomaRt v.2.52.0 (*Durinck et al., 2005*). iCLIP-containing exons were listed once, independently of the number of tags or tag score, and filtered to isolate internal exons using the annotations from Ensembl Genes v.105. Plots were created using ggplot2 v.3.3.6. (*Wickham, 2016*) on R v.4.2.1. Statistical analyses to compare exon length distributions between samples were performed with Wilcoxon Rank Sum and Kruskal-Wallis tests using base R stats package v.4.2.1 and pseudorank v.1.0.1 (*Happ et al., 2020*). Significant enrichment of ultra-long exons in RBMX-regulated and bound exons was tested using the 'N-1' Chi-squared test (*Campbell, 2007*) on the MedCalc Software version 20.218.

## Gene ontology analyses

Gene ontology analyses were performed in R v.4.2.1 using GOstats v.2.62.0 (*Falcon and Gentleman, 2007*) except for [*Figure 3—figure supplement 1A*] for which clusterProfiler::enrichGO v.4.4.4 (*Yu et al., 2012*) was used. Entrez annotations were obtained with biomaRt v.2.52.0 (*Durinck et al., 2005*). Read counts from control treated HEK293 cells (*Liu et al., 2017*) were used to isolate genes expressed in HEK293 cells. Gene ontology analyses for *Figure 3A* were performed for ultra-long (>1000 bp) exons bound or regulated by RBMX against all genes expressed in HEK293 cells that contain ultra-long exons. The Bioconductor annotation data package org.Hs.eg.db v.3.15.0 was used as background for GOBP terms. p-values were adjusted by false discovery rate using the base R stats package v.4.2.1, except for [*Figure 3—figure supplement 1A*] for which the default Benjamini-Hochberg method was used while running enrichGO. Significantly enriched GOBP pathways were

filtered with a p-value cut-off of 0.05. Redundant terms identified with GOstats were removed using Revigo (*Supek et al., 2011*) with SimRel similarity measure against human genes eliminating terms with dispensability score above 0.5. The dot plots were produced using ggplot2 v.3.3.6 (*Wickham, 2016*) focussing on representative terms associated with at least 5% of the initial gene list. Full GOBP lists can be found in *Figure 3—source data 1*.

## Comet assay

The comet assay was performed using the Abcam Comet Assay kit (ab238544) according to the manufacturer's instructions. Briefly U-2 OS cells transfected with RBMX siRNA or control siRNA were harvested after 72 hr, and $1 \times 10^5$ cells were mixed with cold PBS. Cells in PBS were mixed with low-melting comet agarose (1/10) and layered on the glass slides pre-coated with low-melting comet agarose. The slides were lysed in 1 X lysis buffer (pH10.0, Abcam Comet Assay kit) for 48 hr at 4 °C, immersed in Alkaline solution (300 mM NaOH, pH >13, 1 mM EDTA) for 30 min at 4 °C in the dark and then electrophoresed in Alkaline Electrophoresis Solution (300 mM NaOH, pH >13, 1 mM EDTA) at 300 mA, 1 volt/cm for 20 min. The slide was then washed in pre-chilled DI H2O for 2 min, fixed in 70% ethanol for 5 min, and stained with 1 X Vista Green DNA Dye (1/10000 in TE Buffer (10 mM Tris, pH 7.5, 1 mM EDTA), Abcam Comet Assay kit) for 15 min and visualised under fluorescence microscopy Zeiss AxioImager (System 3). Comet quantification was performed using OpenComet (*Gyori et al., 2014*).

## RNA extraction and cDNA synthesis for transcript isoform analysis

RNA was extracted using standard TRIzol RNA extraction (Invitrogen, #15596026) following the manufacturer's instructions. cDNA was synthesised from 500 ng total RNA using SuperScript VILO cDNA synthesis kit (Invitrogen #11754050) following the manufacturer's instructions. To analyse the splicing profiles of the alternative events primers were designed using Primer 3 Plus (*Untergasser et al., 2012*), and the predicted PCR products were confirmed using the UCSC In-Silico PCR tool. *ETAA1* transcript isoform containing the long exon 5 was amplified by RT-PCR using primers 5′-GCTG GACATGTGGATTGGTG-3′ and 5′-GTGCTCCAAAAAGCCTCTGG-3′, while *ETAA1* transcript isoform containing the short exon 5 was amplified using primers 5′-GCTGGACATGTGGATTGGTG-3′ and 5′-GTGGGAGCTGCATTTACAGATG-3′. RT-PCR with this second primer pair could in principle amplify also a 2313 bp product from the *ETAA1* transcript isoform containing the long exon 5, however, PCR conditions were chosen to selectively analyse shorter fragments. *REV3L* 5′-TCACTGTGCAGA AATACCCAC-3′, 5′-AGGCCACGTCTACAAGTTCA-3′, 5′-ACATGGGAAGAAAGGGCACT-3′. *ATRX* 5′-TGAAACTTCATTTTCAACCAAATGCTC-3′ and 5′-ATCAAGGGGATGGCAGCAG-3′ All PCR reactions were performed using GoTaq G2 DNA polymerase kit from Promega following the manufacturer's instructions. All PCR products were examined using the QIAxcel capillary electrophoresis system 100 (Qiagen). Statistical analyses were performed using GraphPad Prism 9.5.0.

## Western blot analyses

Harvested cells treated with either control siRNA or siRNA against RBMX were resuspended in 100 mM Tris-HCL, 200 mM DTT, 4% SDS, 20% Glycerol, 0.2% Bromophenol blue, then sonicated (Sanyo Soniprep 150) and heated to 95 °C for 5 min. Protein separation was performed by SDS-PAGE. Proteins were then transferred to a nitrocellulose membrane, incubated in blocking buffer (5% Milk in 2.5% TBS-T), and stained with primary antibodies diluted in blocking buffer to the concentrations indicated below, at 4 °C overnight. After incubation, the membranes were washed three times with TBS-T and incubated with the secondary antibodies for 1 hr at room temperature. Detection was carried out using the Clarity Western ECL Substrate (Cytiva, RPN2232) and developed using medical X-ray film blue film in an X-ray film processor developer. The following primary antibodies were used at the concentrations indicated: anti-RBMX (Cell Signalling, D7C2V) diluted 1:1000, anti-ETAA1 (Sigma, HPA035048) diluted 1:1000, anti-Tubulin (Abcam, ab18251) diluted 1:2000, anti-GAPDH (Abcepta, P04406) diluted 1:2000, and anti-FLAG (Sigma, F1804) diluted 1:2000.

## Minigene construction and validation

A genomic region containing *ETAA1* exon 5 and flanking intronic sequences were PCR amplified from human genomic DNA using the primers 5′-AAAAAAAAACAATTGAGTTAAGACTTTTCAGCTTTT

CTGA-3′ and 5′-AAAAAAAAACAATTGAGTGCTGGGAAAGAATTCAATGT-3′ and cloned into pXJ41 (*Bourgeois et al., 1999*). Splicing patterns were monitored after transfection into HEK293 cells. RNA was extracted with TRIzol (Invitrogen, #15596026) and analysed using a One Step RT-PCR kit (Qiagen, #210210) following manufacturer's instructions. RT–PCR experiments used 100 ng of RNA in a 5 µl reaction using a multiplex RT-PCR using primers: 5′-GCTGGACATGTGGATTGGTG-3′, 5′-GTGGGAGC TGCATTTACAGATG-3′ and 5′-GTGCTCCAAAAAGCCTCTGG-3′. Reactions were analysed and quantified using the QIAxcel capillary electrophoresis system 100 (Qiagen).

## Branchpoint analysis

Using the RNA from the long minigene transfections with RBMX and RBMXΔRRM, total RNA was extracted using TRIZOL reagent (Life Technologies) following the standard manufacturer's instructions, RNA concentration was quantified by NanoDrop UV-Vis spectrophotometer and treated with DNase I (Invitrogen, Am1906). 1 µg of purified RNA was reverse transcribed with a SuperScript III Reverse Transcriptase (Invitrogen, #18080093) using ETAA1 DBR R1 RT-PCR primer 5′-AAGTTCTTCTTCTTGA CTTTGTGTT-3′ and treated with RNaseH (New England Biolabs, M0297S). 1 µl of the cDNA was used for PCR amplification reactions using using GoTaq G2 DNA Polymerase (Promega, #M7845) in the standard 25 µl reaction following the manufacturer's instructions. PCR amplification was carried out using two different primer sets ETAA1 DBR R2 5′-GCTCTTGAATCACATCTAGCTCT-3′ and ETAA1 DBR F1 5′-AGCCAAACTAACTCAGCAACA-3′, ETAA1 DBR R2 5′-GCTCTTGAATCACATCTAGCTCT-3′ and ETAA1 DBR F2 5′-AGCATTTGAATCCAGGCAGC-3′ with 18 cycles of amplification at an annealing temperature of 56 °C. PCR products were sub-cloned into the pGEM-T-Easy vector (Promega, A1360) using manufacture instructions. Plasmids were subjected to Sanger sequencing (Source BioScience) and the sequences were checked with BioEdit (*Hall, 1999*) and aligned to the genome with UCSC (*Karolchik et al., 2014*).

## Acknowledgements

This work was supported by the Biotechnology and Biological Sciences Research Council (BBSRC) (grants BB/S008039/1, BBW002019/1, and BB/P006612/1), and by the JGW Patterson Foundation. Saad Aldalaqan was supported by a PhD studentship from the King Fahad Medical City. Chileleko Siachisumo was supported by the Newcastle Liverpool Durham BBSRC Doctoral Training Partnership (BB/M011186/1).

## Additional information

### Funding

| Funder | Grant reference number | Author |
| --- | --- | --- |
| Biotechnology and Biological Sciences Research Council | BB/S008039/1 | Caroline Dalgliesh David J Elliott |
| Biotechnology and Biological Sciences Research Council | BB/W002019/1 | Sara Luzzi Caroline Dalgliesh Ingrid E Ehrmann David J Elliott |
| Biotechnology and Biological Sciences Research Council | Newcastle Liverpool Durham BBSRC Doctoral Training Partnership (BB/M011186/1) | Chileleko Siachisumo |
| Biotechnology and Biological Sciences Research Council | BB/P006612/1 | Ingrid E Ehrmann David J Elliott |
| King Fahad Medical City | | Saad Aldalaqan |

The funders had no role in study design, data collection and interpretation, or the decision to submit the work for publication.

## Author contributions
Chileleko Siachisumo, Investigation, Writing – original draft; Sara Luzzi, Conceptualization, Funding acquisition, Investigation, Writing – original draft; Saad Aldalaqan, Gerald Hysenaj, Investigation; Caroline Dalgliesh, Matthew R Gazzara, Ivaylo D Yonchev, Ingrid E Ehrmann, Investigation, Writing – review and editing; Kathleen Cheung, Katherine James, Simon J Cockell, Formal analysis; Mahsa Kheirollahi Chadegani, Resources, Methodology; Graham R Smith, Data curation, Writing – review and editing; Jennifer Munkley, Supervision, Investigation; Stuart A Wilson, Supervision, Writing – review and editing; Yoseph Barash, Supervision, Investigation, Writing – review and editing; David J Elliott, Conceptualization, Supervision, Funding acquisition, Investigation, Project administration, Writing – review and editing, Writing – original draft

## Author ORCIDs
Sara Luzzi ⓘD https://orcid.org/0000-0001-6337-7495
Simon J Cockell ⓘD https://orcid.org/0000-0002-6831-9806
David J Elliott ⓘD https://orcid.org/0000-0002-6930-0699

Reviewer #1 (Public Review): https://doi.org/10.7554/eLife.89705.3.sa1
Reviewer #2 (Public Review): https://doi.org/10.7554/eLife.89705.3.sa2
Reviewer #3 (Public Review): https://doi.org/10.7554/eLife.89705.3.sa3
Author response https://doi.org/10.7554/eLife.89705.3.sa4

---

# Additional files

## Supplementary files
• MDAR checklist

## Data availability
The data discussed in this publication have been deposited in NCBI's Gene Expression Omnibus (*Edgar et al., 2002*) and are accessible through GEO series accession number GSE233498. Previously published GSE74085 data was also accessed from Gene Expression Omnibus.

The following dataset was generated:

| Author(s) | Year | Dataset title | Dataset URL | Database and Identifier |
|---|---|---|---|---|
| Smith G | 2023 | An anciently diverged family of RNA binding proteins maintain correct splicing of ultra-long exons through cryptic splice site repression | https://www.ncbi.nlm.nih.gov/geo/query/acc.cgi?acc=GSE233498 | NCBI Gene Expression Omnibus, GSE233498 |

The following previously published dataset was used:

| Author(s) | Year | Dataset title | Dataset URL | Database and Identifier |
|---|---|---|---|---|
| Liu N, Parisien M, Dai Q, Zhou K, Diatchenko L, Pan T | 2017 | N6-methyladenosine alters RNA structure to regulate binding of a low-complexity protein | https://www.ncbi.nlm.nih.gov/geo/query/acc.cgi?acc=GSE74085 | NCBI Gene Expression Omnibus, GSE74085 |

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
