## [Editor Report · eLife assessment]

This **important** paper addresses the process by which cryptic splice sites that occur randomly in exons are ignored by the splicing machinery. Integrating state-of- the-art genome-wide approaches such as CLIP-seq with the study of individual examples, this study **convincingly** implicates members of RBMX family of RNA binding proteins in such cryptic splice site suppression and showcases its importance for the fidelity of expression of genes with very large exons.

---

## [Referee Report · Reviewer #1 (Public Review)]

Summary:

The article by Siachisumo, Luzzi and Aldalaquan et al. describes studies of RBMX and its role in maintaining proper splicing of ultra-long exons. They combine CLIP, RNA-seq, and individual example validations with manipulation of RBMX and its family members RBMY and RBMXL2 to show that the RBMX family plays a key role in maintaining proper splicing of these exons.

I think one of the main strengths of the manuscript is its ability to explore a unique but interesting question (splicing of ultra-long exons), and derive a relatively simple model from the resulting genomics data. The results shown are quite clean, suggesting that RBMX plays an important role in proper regulation of these exons. The ability of family members to rescue this phenotype (as well as only particular domains) is also quite intriguing and suggests that the mechanisms for keeping these exons properly spliced may be a quite important and highly conserved mechanism.

The revised manuscript addresses many of my earlier critiques and does an effective job of arguing that RBMX plays a large-scale role in regulating splicing of long exons. I think there are obvious open questions for future work (the mechanism of how RBMX/RBMXL2 achieve this splicing control is perhaps hinted at but not fully explored here), but I think the article provides an intriguing analysis of the role of RBMX that will activate interesting future studies.

---

## [Referee Report · Reviewer #2 (Public Review)]

Summary:

One of the greatest challenges for the spliceosome is to be able to repress the many cryptic splice sites that can occur in both the intronic and exotic sequences of genes. Although many studies have focused on cryptic signals in introns (because of their common involvement in disease) the question still remained open as to the factors that repress cryptic exons in exons. Because exons are normally much shorter than introns, in many cases the problem does not exist. However, in human genes a significant proportion of exons can be considerably longer than the average 150 nt length and this raises the question of how cryptic splicing can be prevented in long exons. To address this question, the authors have focused on the possible role played by an ancient mammalian RBD protein called RBMX. Using a combination of high-throughput and classic splicing methodologies, they have shown that there is a class of RBMX-dependent ultra-long exons connected where the RBMX, RBMXL2 and RBMY paralogs have closely related functional activity in repressing cryptic splice site selection.

Strengths:

In general, the present work sheds light on what has been a rather understudied process in splicing research. The use of iCLIP and RNA-seq data has not only allowed to identify the long exons where cryptic splicing is prevented by the RBMX proteins but has also allowed to identify a network of genes mostly involved in genome stability and transcriptional control where these proteins seem to play a prominent role. This can therefore also shed additional information on the way splicing has shaped evolutionary processes in the mammalian lineage and will therefore be of interest to many researchers in this field.

Weaknesses:

There are no major weaknesses, although some specific aspects of the findings could be addressed more in-depth in the recommendations to authors.

---

## [Referee Report · Reviewer #3 (Public Review)]

The manuscript by Siachisumo et al builds upon a previous publication from the same group of collaborators that showed that depletion of mouse RBMXL2 leads to a block in spermatogenesis associated with mis-splicing, particularly of large exons in genes associated with genome stability (Ehrmann et al Elife 2019). RBMXL2 is an RNA-binding protein and an autosomal retrotransposed paralog of the X-chromosomally encoded RBMX. RBMXL2 is expressed during meiosis when RBMX and the more distantly related RBMY (on the Y chromosome) are silenced. It is therefore an appealing hypothesis that RBMXL2 might provide cover for RBMX function during meiosis. To address this hypothesis the authors analysed the transcriptomic consequences of RBMX depletion by RNA-Seq in human cells (MDA-MB-231 and existing RNA-Seq data from HEK293 cells), complemented by iCLIP to analyze the binding targets of FLAG-tagged RBMX in HEK293 cells. The findings convincingly demonstrate that - like RBMXL2 - RBMX mainly acts as a splicing repressor and that it particularly acts to protect the integrity of very long ("ultra-long") exons, defined as those over 1000 nt. Upon RBMX depletion, many of these exons are shortened due to the use of cryptic 5' and/or 3' splice sites. Moreover, affected genes are particularly enriched for functions associated with genome integrity - indeed "comet assays" show that RBMX depletion leads to DNA damage defects. Strikingly, RNA-Seq analysis showed that overexpression of RBMXL2 is able to complement the majority of splicing changes caused by RBMX depletion, particularly those involving ultra-long exons. In a smaller scale experiment RBMY was also able to complement effects of RBMX knockdown upon three target events in the ETAA1, REV3L and ATRX genes.

In addition to these core findings the manuscript also includes some experiments that begin to address more mechanistic questions, such as the potential for RBMX to sterically block access of spliceosome components to splice site elements, and preliminary structure-function analyses of RBMX showing that its RRM domain is not necessary for splicing regulatory activity on the ETAA1, REV3L and ATRX target events.

In summary, this manuscript provides clear and convincing evidence to support the role of RBMX in somatic cells as a repressor of cryptic splice sites in ultra-long exons, mirroring the function of RBMXL2 in meiotic cells. It therefore demonstrates how the RBMX/RBMXL2/RBMY family perform a key role in protecting the transcriptomic integrity of ultra-long exons.

---

## [Author Response]

The following is the authors’ response to the original reviews.

(1) The conclusions in the text are very broad and general but often based on a limited number of examples. It would be important that the authors hit the appropriate tone when most of the analysis (in Figure 5) is derived from n=3 events.

We have tried to hit the correct tone here by modifying our manuscript text. In particular we have we have added a pie chart to Figure 4 (Figure 4C, that summarises data from all RBMX targets, not just the original n=3, and shows that most RBMX targets are rescued by RBMXL2).

(2) The fractions of long/ultra-long exons actually bound by/regulated by RBMX are not clearly stated - which is in contrast to the general statement of the title (implying a global role for RBMX in proper splicing of ultra-long exons).

(i) We have changed our title (now “An anciently diverged family of RNA binding proteins maintain correct splicing of a class of ultra-long exons through cryptic splice site repression”).

(ii) We also include much more clear text about the fractions of long/ultralong exons bound by RBMX with the following text:

“…..This led us to test whether RBMX protein is preferentially associated with long exons. For this we plotted the distribution of internal exons bound and regulated by RBMX together with all internal exons expressed from HEK293 mRNA genes (Liu et al., 2017) (Figure 2 – Source Data 1). We found that RBMX controls and binds two different classes of exons: the first have comparable length to the average HEK293 exon, while the second were extremely long, exceeding 1000 bp in length (Figure 2F). We defined this second class as ‘ultra-long exons’, which represented the 18.9% of internal exons regulated by RBMX and 17.6% of the ones that contained RBMX iCLIP tags. These proportions were significantly enriched compared to the general abundance of internal ultra-long exons expressed from HEK293 cells, which was only 0.4% (Figure 2G)……”

“…….We next wondered whether ultra-long exons regulated by RBMX (which represented 11.6% of all ultra-long internal exons from genes expressed in HEK293) had any particular feature compared to ultra-long exons that were RBMX-independent……..”

(3) The authors should state what fraction of ultra-long exons show cryptic splicing in the RBMX siRNA that are corrected by RBMXL2 overexpression (rather than just showing the 3 events). There's some confusion about the global nature of the conclusions relative to the data displayed.

This is a good point. We have used the RNAseq information as suggested, and included a pie chart (Figure 4C) that includes this information.

(4) It would be helpful if the authors could identify if there are some motifs more present in ultra-long exons than others.

Good point, we have included k-mer analysis of the ultra-long exons bound by RBMX, and also more generally ultra-long exons in the human genome, in Figure 2H and 2I. We also add the following text:

K-mer analyses also showed that while ultra-long exons within mRNAs are rich in AT-rich sequences compared to shorter exons (Figure 2H), the ultra-long exons that are either regulated or bound by RBMX displayed enrichment of AG-rich sequences (Figure 2I), consistent with our identified RBMX-recognised sequences (Figure 2C).

(5) The authors should evaluate if RBMX-repressed 3' splice sites have similar or low splice site scores/strengths than natural 3' splice sites.

We have added splice site score analyses in Figure 1F and Figure 1 Supplement 1B. These show that the cryptic splice sites repressed by RBMX are not significantly different from those that are normally used. We add the following text to accompany these figure panels:

“Furthermore, analysis of splice site strength revealed that, unlike splice sites activated by RBMX (Figure 1 – Figure supplement 1B), alternative splice sites repressed by RBMX have comparable strength to more commonly used splice sites (Figure 1F). This means that RBMX operates as a splicing repressor in human somatic cells to prevent use of ‘decoy’ splice sites that could disrupt normal patterns of gene expression.”

(6) The section "RBMX protein-RNA interactions may insulate important splicing signals from the spliceosome." is a very preliminary look at possible mechanisms. Can you integrate the RNA Seq and CLIP datasets to generate "splicing maps" that would provide more generalized insights? In fact, where possible, it would be great to integrate the iCLIP data from the same cell types to generate RNA splicing maps (with the KD RNA-seq data)

We have added “RNA map-type” plots to integrate iCLIP data with splicing patterns (Figure 2 Figure supplement 1D and 1E), and made corresponding changes to the text.

Additional changes

We also made some extra changes to respond to the further points raised by reviewers.

(1) We have carried out gene ontology analysis of those genes that contain RBMX-regulated ultra-long exons versus all ultra-long exons (now Figure 3A, and also Figure 3- Figure supplement 1A and 1B).

(2) We have corrected the cartoon summarising the branch point analysis (now Figure 3 – Figure Supplement 2F).